# Combined Score of Perivascular Space Dilatation and White Matter Hyperintensities in Patients with Normal Cognition, Mild Cognitive Impairment, and Dementia

**DOI:** 10.3390/medicina58070887

**Published:** 2022-07-01

**Authors:** Nauris Zdanovskis, Ardis Platkājis, Andrejs Kostiks, Kristīne Šneidere, Ainārs Stepens, Roberts Naglis, Guntis Karelis

**Affiliations:** 1Department of Radiology, Riga Stradins University, LV-1007 Riga, Latvia; ardis.platkajis@rsu.lv; 2Department of Radiology, Riga East University Hospital, LV-1038 Riga, Latvia; rob.naglis@gmail.com; 3Military Medicine Research and Study Centre, Riga Stradins University, LV-1007 Riga, Latvia; kristine.sneidere@rsu.lv (K.Š.); ainars.stepens@rsu.lv (A.S.); 4Department of Neurology and Neurosurgery, Riga East University Hospital, LV-1038 Riga, Latvia; andrejs.kostiks@gmail.com (A.K.); guntis.karelis@rsu.lv (G.K.); 5Department of Health Psychology and Paedagogy, Riga Stradins University, LV-1007 Riga, Latvia; 6Department of Infectology, Riga Stradins University, LV-1007 Riga, Latvia

**Keywords:** dementia, cognitive impairment, MRI, brain, perivascular spaces, white matter hyperintensities

## Abstract

*Background and Objectives*: Cerebral perivascular spaces (PVS) are part of the cerebral microvascular structure and play a role in lymphatic drainage and the removal of waste products from the brain. White matter hyperintensities (WMH) are hyperintense lesions on magnetic resonance imaging that are associated with cognitive impairment, dementia, and cerebral vascular disease. WMH and PVS are direct and indirect imaging biomarkers of cerebral microvascular integrity and health. In our research, we evaluated WMH and PVS enlargement in patients with normal cognition (NC), mild cognitive impairment (MCI), and dementia (D). *Materials and Methods*: In total, 57 participants were included in the study and divided into groups based on neurological evaluation and Montreal Cognitive Assessment results (NC group 16 participants, MCI group 29 participants, D group 12 participants). All participants underwent 3T magnetic resonance imaging. PVS were evaluated in the basal ganglia, centrum semiovale, and midbrain. WMHs were evaluated based on the Fazekas scale and the division between deep white matter (DWM) and periventricular white matter (PVWM). The combined score based on PVS and WMH was evaluated and correlated with the results of the MoCA. *Results:* We found statistically significant differences between groups on several measures. Centrum semiovale PVS dilatation was more severe in MCI and dementia group and statistically significant differences were found between D-MCI and D-NC pairs. PVWM was more severe in patients with MCI and dementia group, and statistically significant differences were found between D-MCI and D-NC pairs. Furthermore, we found statistically significant differences between the groups by analyzing the combined score of PVS dilatation and WMH. We did not find statistically significant differences between the groups in PVS dilation of the basal ganglia and midbrain and DWM hyperintensities. *Conclusions:* PVS assessment could become one of neuroimaging biomarkers for patients with cognitive decline. Furthermore, the combined score of WMH and PVS dilatation could facilitate diagnostics of cognitive impairment, but more research is needed with a larger cohort to determine the use of PVS dilatation and the combined score.

## 1. Introduction

Neurodegenerative diseases represent a big burden on today’s society, and it is projected that dementia will have significant societal and economic implications in the future [1,2]. Therefore, it is important to explore new radiological biomarkers to improve the early diagnosis of cognitive impairment with radiological methods.

The cerebral microvascular structure consists of several anatomical structures that could be identified on magnetic resonance (MR) scanning (i.e., cerebral perivascular spaces (PVS) or Virchow-Robin spaces) [3]. Additionally, on MR we can see lesions that could serve as direct and indirect biomarkers of microvascular integrity. That is, periventricular white matter hyperintensities and deep white matter hyperintensities.

***Cerebral perivascular spaces*** (PVS or Virchow-Robin spaces) are part of the cerebral microvascular structure, they are surrounded by the adventitia of the vessel and astrocyte end-feet as they course from subarachnoid space through the brain parenchyma [4,5,6]. The interstitial fluid within the brain parenchyma drains from the gray matter of the brain and flows along with PVS, that is, it works as the lymphatic drainage of the brain and removes waste products such as amyloid beta, which is known to play a role in the pathogenesis of Alzheimer’s disease [4,7,8,9]. Additionally, the presence of PVS is associated with increased amyloid-β (Aβ) deposition in the leptomeningeal arteries [10].

Perivascular spaces are divided into four subtypes: *Type 1*—Located in basal ganglia along perforating lenticulostriate arteries; *Type 2*—Located in centrum semiovale along the path of perforating medullary arteries in high convexities and white matter; *Type 3*—Located in the midbrain at the pontomesencehpalic junction; *Type 4*—recently there is a proposition to include 4th type that includes anterior temporal PVS dilatation [4,11,12,13,14].

Currently, the clinical relevance of enlargement of perivascular spaces is a matter of discussion, and there are ambiguous results regarding the relevance of PVS in dementia [15]. There are research articles where it shows a connection with small vessel disease [16,17,18], cognitive impairment, and dementia [19,20]. Furthermore, a large meta-analysis indicates the need for further studies to determine PVS with different neurological disorders, including cognitive impairment and dementia [21].

***White matter hyperintensities*** are hyperintense lesions on MRI T2 FLAIR imaging and are a well-known risk factor for vascular dementia and increased risk of cerebrovascular events. The most used grading scale for white matter lesions is the Fazekas scale. White matter hyperintensities are a combination of axon degeneration of axons and myelin, gliosis, and small vessel ischemia [22]. Furthermore, WMH could predict amyloid increase in amyloids in Alzheimer’s disease and thus is an important radiological biomarker for assessing patients with cognitive impairment [23]. In addition, the presence and progression of white matter hyperintensities could have a more global effect on cognitive function [24].

On the Fazekas scale, there is a distinction between periventricular white-matter (PVWM) and deep white-matter (DWM) lesions [25,26,27]. There are histopathological, functional and etiological differences between DWM and PVWM, that is, PVWM lesions are reported to have a connection with cognitive impairment and DWMH are more associated with mood disorders [26,27,28,29], although it is worth mentioning that some research shows the opposite effect and indicates the association of PVWM lesion with cognitive impairment [30]; postmortem histopathological studies indicate that PVWM have discontinued ependyma, gliosis, loss of white matter fibers and myelin loss, and DWM shows more axonal loss, infarction, demyelination and gliosis [31,32].

While WMH and PVS are important biomarkers of cerebral microvascular health, other magnetic resonance imaging that could indicate cerebral microvascular disease are small subcortical infarctions, cerebral microbleeds, and lacunes. In general, WMH and PVS are important direct and indirect biomarkers of cerebral microvascular health and have an impact on each other [33,34,35,36].

PVS dilatation and WMH could be used as biomarkers for cerebral microvascular disease. At the moment, PVS evaluation is not widely used, nor endorsed in clinical practice to evaluate patients with cognitive impairment. In our research work, we analyzed PVS dilatation, WMH and we also used combined score of PVS dilatation and WMH in participants without cognitive impairment, mild cognitive impairment, and dementia.

## 2. Materials and Methods

This is a prospective cross-sectional study that includes a patient’s cognitive assessment and subsequent MRI scan.

### 2.1. Participant Selection

In our study participants were admitted to the neurologist with subjective cognitive impairment or suspected cognitive impairment. The neurologist is board certified neurologist that specializes in cognitive impairment diagnostics and management. Exclusion criteria for participants were clinically significant neurological diseases (tumors, major stroke, intracerebral lobar hemorrhages, malformations, Parkinson’s disease, multiple sclerosis, etc.), substance abuse, and alcohol abuse. Patients with clinically significant prior know vascular disease were not included in our study. Study participants did not have other significant pathological findings on magnetic resonance imaging (MRI).

Detailed participant demographic data and cognitive assessment scores are available in results section.

### 2.2. Magnetic Resonance Image (MRI) Acquisition and Sequences

MRI scans were performed on a 3.0 tesla General Electric (GE) MRI scanner in a university hospital setting. PVS were analyzed in T2 axial sequences (TR (repetition time) 4671, TE (time to echo) 130.6, slice thickness 3 mm) and white matter hyperintensities were analyzed on 3D T2 FLAIR (fluid-attenuated inversion recovery) axial sequences (TR 6002.0, TE 136.6, slice thickness 2 mm). Additionally, Ax T2 3D SWAN (susceptibility-weighted angiography), 3D Ax T1, and DWI (diffusion weighted imaging) sequences were analyzed to exclude other clinically significant pathologies.

### 2.3. Perivascular Space Grading Scale

Perivascular space grading was partially based on Potter et al. [13] grading scale. We evaluated basal ganglia PVS (Type 1, see Figure 1) and centrum semiovale PVS (Type 2, see Figure 2) by the rating scale: grade 0–no PVS dilatation; Grade 1–1 to 10 dilatated PVS; Grade 2–11 to 20 dilatated PVS; Grade 3–21 to 40 dilatated PVS; Grade 4–>40 dilatated PVS.

Midbrain PVS (Type 3) were evaluated whether PVS are visible (Grade 1) or there is no PVS dilatation (Grade 0).

In our participants, we did not find Type 4 PVS dilatation.

### 2.4. DWM and PVWM Hyperintensity Grading Scales

We used Fazekas scale grading by labeling patient T2 FLAIR images from Grade 1 to Grade 3 and divided lesion location in periventricular white-matter lesions and deep white-matter lesions.

In periventricular white matter lesions Grade 1 was assigned to patients with mild hyperintensities that occurred along with the ventricular horns, also referred to as ventricular caps, Grade 2 was assigned to patients with more extensive hyperintensities and Grade 3 was assigned to patients with lesions that were extending into deep white matter (see Figure 3).

Deep white matter hyperintensity Grade 1 was assigned to participants who had small, punctate foci in subcortical regions, Grade 2 was assigned to participants who had a higher number and more extensive foci, and Grade 3 was assigned to patients with confluent white matter hyperintensities (see Figure 4).

### 2.5. Combined Score

To evaluate the combined effects of perivascular space dilatation and white matter hyperintensities we used a combined score of the lesions, i.e., the patient who has Grade 3 PVS dilatation in the basal ganglia (3 points), Grade 3 PVS dilatation in centrum semiovale (3 points), PVS space dilatation in the midbrain (1 point), and Fazekas scale Grade 3 lesions in periventricular regions (3 points) and Fazekas scale Grade 3 lesions in subcortical regions (3 points) in total gets 13 points. Patients without lesions get 0 points. Therefore, the higher the combined score, the more severe the microvascular structure lesions are present.

### 2.6. Statistical Analysis

Statistical analysis was performed using the freeware software JASP 0.16.0. (Eric-Jan Wagenmakers, Amsterdam, The Netherlands) [37]. Statistical analysis included descriptive statistics, the Kruskal-Wallis test, and post hoc analysis of our results. A Spearman’s correlation was run to determine the relationship between MoCA test results and combined score as well as basal ganglia PVS grade, centrum semiovale PVS grade, midbrain PVS grade, PVWM, and DWM.

## 3. Results

### 3.1. Participants and Cognitive Assessment

Participants were admitted to the neurologist that is specialized in cognitive impairment diagnostics and underwent the Montreal Cognitive Assessment (MoCA). Based on neurological assessment and MoCA scores, patients were divided into three groups [38,39,40]:Participants with normal cognition (NC) with MoCA scores ≥ 26,Participants with mild cognitive impairment (MCI) with MoCA scores ≥ 18 and ≤25,Participants with dementia (D) with MoCA scores ≤ 17.

All participants had at least 16 years of higher education.

In total, we included 57 participants–in the normal cognition group 16 participants (mean age 62.6, SD 15.4, youngest patient 62, oldest patient 96, mean MoCA score 28.1, SD 1.3, lowest score 26, highest score 30), in the MCI group 29 patients (mean age 72.5, SD 7.1, youngest patient 57, oldest patient 85, mean MoCA score 22.4, SD 2.4, lowest score 18, highest score 25) and in the dementia group 12 patients (mean age 73.3, SD 10.2, youngest patient 62, oldest patient 96, mean MoCA score 9.3, SD 4.2, lowest score 4, highest score 15). Demographic data, gender, and MoCA scores of participants between groups can be seen in Table 1.

Pearson’s Chi-Square test on gender was performed and there were no statistically significant differences between the groups (χ^2^ = 2.291, *p* = 0.318).

A Kruskal-Wallis H test was conducted to examine the age differences between groups and no significant differences were found between the groups (χ^2^ = 2.633, *p* = 0.268).

A Kruskal-Wallis H test was conducted to examine the differences in MoCA score between groups and there were statistically significant differences between the groups (χ^2^ = 48.016, *p* < 0.001). Based on neurological assessment and magnetic resonance imaging data, participants in the MCI and D groups were not exclusively classified as one type of dementia (i.e., vascular dementia, Alzheimer’s disease, Lewy body dementia, or frontotemporal dementia). All patients in the MCI or D groups had at least some T2 white matter hyperintensities and some degree of global cortical atrophy in combination with other cerebral lobe atrophies.

### 3.2. PVS Dilatation in Basal Ganglia, Centrum Semiovale, and Midbrain

***Basal ganglia PVS*** Grade 3 and Grade 4 dilatation was encountered more often in patients with dementia (see distribution in Table 2 and box plot in Figure 5).

In the dementia group Grade 3 and Grade 4 basal ganglia PVS dilatation was observed in 33% of cases (4 out of 12 participants), in MCI group 22% of cases (6 out of 27 participants), and 0% in the normal cognition group.

By performing the Kruskal-Wallis test we did not find statistically significant differences between dementia, MCI, and NC groups.

***Centrum semiovale PVS*** dilatation was encountered more often in patients with dementia (see distribution in Table 3 and boxplot in Figure 6).

In the dementia group grade 3 and grade 4 centrum semiovale PVS dilatation was seen in 42% of cases (5 out of 12 participants), in MCI group 7% of cases (2 out of 27 participants), and 0% in the normal cognition group.

By performing the Kruskal-Wallis test we found statistically significant differences between the groups (H (2) = 6.387, *p* < 0.05). Pairwise comparisons showed that there are statistically significant differences between dementia–MCI groups (*p* < 0.05) and dementia-NC groups (*p* < 0.01). Post hoc testing using the Bonferroni and Holm-Bonferroni correction revealed statistically significant differences in same groups. There were no statistically significant differences between MCI-NC groups (see Table 4).

PVS dilatation of the midbrain was encountered more frequently in patients with dementia than in other groups. From the Dementia group, 75% (9 out of 12) participants had dilatation of midbrain PVS, in the MCI group 59% (16 out of 27) participants had dilated PVS, in normal cognition 44% (8 out of 18) participants had dilated PVS.

A chi-square test of independence examined the relationship between groups (NC, MCI, D) and midbrain perivascular space dilatation (dilated, non-dilated). The relationship between these variables was not statistically significant, χ^2^ (2, N = 57) = 1.421, *p* = 0.233. We did not find statistically significant differences between dementia, MCI, and NC groups.

### 3.3. White Matter Hyperintensities

In the dementia group, **PVWM hyperintensities** had higher grade according to Fazekas scale and higher grade was encountered more often in the dementia group (see distribution in Table 5 and boxplot in Figure 7).

In the dementia group, 33% (4 out of the 12) of participants had severe Grade 3 PVWM hyperintensities, on the contrary, in the MCI group and in the NC group participants did not have severe PVWM hyperintensities.

By performing the Kruskal-Wallis test we found statistically significant differences between the groups (H (2) = 12.513, *p* < 0.01). Pairwise comparisons showed that there are statistically significant differences between dementia–MCI groups (*p* < 0.01) and dementia-NC groups (*p* < 0.001). Post hoc testing using the Bonferroni and Holm-Bonferroni correction revealed statistically significant differences in same groups. There were no statistically significant differences between MCI-NC groups (see Table 6).

In the dementia group, the average grade of **DWM hyperintensities** had a higher value according to the Fazekas scale and a higher grade was encountered more often in the dementia group than in other groups (see distribution in Table 7 and boxplot in Figure 8).

In the dementia group 50% (6 out of 12) of the participants had severe (Grade 3) or moderate (Grade 2) hyperintensities of DWM, in the MCI group 18% (4 out of 28) and NC group 11% (2 out of 18).

By performing the Kruskal-Wallis test, we did not find statistically significant differences between the dementia, MCI and NC groups.

### 3.4. PVS, WMH and Combined Score Correlation with MoCA

To summarize our findings for microvascular lesions, we calculated the combined score for participants where the highest possible score was 13 and the lowest possible score 0. The highest combined score could be achieved if the patient has grade 3 PVS dilatation in basal ganglia (3 points), Grade 3 PVS dilatation in centrum semiovale (3 points), PVS space dilatation in midbrain (1 point), Fazekas scale grade 3 lesions in periventricular regions (3 points) and Fazekas scale grade 3 lesions in subcortical regions (3 points).

We found that the combined score was higher in patients with dementia and MCI (see Table 8 and Figure 9).

Performing the Kruskal-Wallis test, we found statistically significant differences between the groups (H (2) = 10.479, *p* < 0.01). Pairwise comparisons showed that there are statistically significant differences between the dementia–MCI groups (*p* < 0.01) and the dementia-NC groups (*p* < 0.001). Post hoc testing using the Bonferroni and Holm-Bonferroni correction revealed statistically significant differences in the same groups. There were no statistically significant differences between the MCI-NC groups (see Table 9).

Spearmen’s rho correlation coefficient was used to assess the relationship between basal ganglia PVS grade, centrum semiovale PVS grade, midbrain PVS grade, PVWM hyperintensity grade, DWM hyperintensity grade, total Fazekas scale grade, combined score and the results of the MoCA test

Combined score Spearmen’s correlation with MoCA showed that there is a statistically significant negative correlation (r = −0.446, *p* < 0.001, see Figure 10). When correcting Spearmen’s correlation for age and gender statistically significant results remained (r = −0.406, *p* = 0.002).

Additionally, we assessed basal ganglia PVS grade, centrum semiovale PVS grade, midbrain PVS grade, PVWM, DWM, correlation with MoCA where we found statistically significant correlation in all variables except midbrain PVS grade (see Table 10).

## 4. Discussion

In general, it is a necessity to identify valuable imaging biomarkers and develop visual rating scales for the evaluation of cognitive impairment, including microvascular biomarkers [41,42].

White matter hyperintensity evaluation is a well-recognized and widely used imaging biomarker. The most commonly used WMH scale is the Fazekas scale and is also recommended by the Imaging Cognitive Impairment Network (ICINET) [25,42]. The histopathology of WMH is a combination of axonal loss and myelin damage that is caused mainly by small vessel disease and ischemia. WMH is associated with increased vascular risk, stroke recurrence, and unfavorable functional outcome after stroke [43,44,45] and decreased cognitive abilities, affecting memory and executive function [22,46,47,48,49].

By analyzing the hyperintensities of DWM and PVWM, we found that there were statistically significant differences in PVWM between our groups and there were no statistically significant differences in the DWM group. Therefore, our results are consistent with *Kim et al., 2008*, and contradict *Bolandzadeh et al.*, *2012.* More studies are needed to confirm our findings.

Contrary to the evaluation of WMH, the PVS evaluation is not widely used nor endorsed in clinical practice to evaluate patients with cognitive impairment. The most often used scales are the medial temporal atrophy scale (MTA), the global cortical atrophy scale (GCA), and white matter hyperintensities [42,50,51]. The consensus regarding PVS is that they play a role in fluid transport, in exchange between cerebrospinal fluid and interstitial fluid, and in the removal of waste products from the brain (including Aβ) [52]

There are research articles that indicate the connection of PVS dilation with small vessel disease and stroke [53,54]; cognitive decline and dementia [15,20].

In our study, we found statistically significant differences between groups in centrum semiovale PVS dilatation and did not find statistically significant differences by analyzing basal ganglia PVS and midbrain PVS. The results from other studies are ambiguous, that is, Valdes Hernandez et al. proposed that the semiovale PVS is not directly associated with cognitive abilities in older age, while Kim et al. found that PVS dilation is associated with Aβ positivity and thus could be an indirect imaging marker of Alzheimer’s dementia [55,56]. A systematic review of enlarged PVS indicates heterogeneity of assessment and that more longitudinal data with multivariate analysis is necessary to define a clear association with dementia [15].

PVS play an important role in the glymphatic system and is part of waste product elimination system. Glymphatic system has been described in detail in several studies [57,58]. Also, glymphatic disfunction has been linked to cognitive decline [59,60].By identifying the connection of PVS with MCI and dementia, there could be potential therapeutic agents that manipulate glymphatic activity to improve waste product clearance from the brain [61].

To evaluate how these imaging biomarkers could be used in conjunction and to summarize our findings, we made a combined score that is a sum of the WMH lesion grades and the PVS dilatation grade. We found statistically significant differences between dementia–normal cognition and dementia–mild cognitive impairment pairs. Additionally, we found a statistically significant negative correlation between MoCA scores and combined scores, that is, a higher combined score is associated with lower MoCA scores. These findings also confirmed our observations, since some participants in the dementia group had exclusively severe PVS dilation and some participants had exclusively severe WMH, thus the combined score shows us a general overview of the use and implication of the PVS grading scale and the WMH grading scale.

Although this was an exploratory study, there are several limitations of our study. The results must be interpreted with caution due to an inhomogeneous study group: The participants in the normal cognition group were younger than the patients in the MCI and dementia group. Considering the limited sample size, a larger cohort is needed to confirm the involvement of the PVS space in the pathogenesis of cognitive impairment. Furthermore, in our study, we did not identify patients with other clinical conditions that could cause PVS dilation, that is, diabetes status and glucose levels status, hypertension, dyslipidemia, smoking habits, obstructive sleep apnea, etc. A downside of the cross-sectional study design used in PVS evaluation is a discussion of what came first-either there is a loss of brain volume that was caused by PVS dilatation, or we see PVS dilation as a result of a brain atrophy itself.

With this study, our objective was to explore the differences in WMH and PVS between groups of normal cognition, MCI, and dementia. Additionally, we assessed the combined score of WMH and PVS dilatation between the groups, and we assessed the correlation of the MoCA result with combined score, WMH and PVS dilatation.

PVS has the potential to become radiological biomarkers for patients with cognitive impairment. To confirm our findings, it is necessary to validate the results in the larger cohort of patients and include more longitudinal data, including a focus on comorbidities that could cause PVS dilation.

## 5. Conclusions

The data presented in our study showed statistically significant differences of PVS dilatation in the centrum semiovale between D-MCI and D-NC study groups, and statistically significant differences of PVWM hyperintensities between D-MCI and D-NC study groups. In contrast, we did not find statistically significant differences between the groups in the PVS dilatation in basal ganglia and the midbrain, and DWM hyperintensities. The combined PVS and WMH showed statistically significant differences between the study groups and had a statistically significant correlation with MoCA results.

The PVS assessment could become one of the neuroimaging biomarkers for patients with cognitive decline. Furthermore, the combined WMH and PVS dilatation could facilitate the diagnostics of cognitive impairment, but more research is needed with a larger cohort to determine the use of PVS dilation and the combined score as an imaging biomarker in patients with cognitive impairment.

## Figures and Tables

**Figure 1 medicina-58-00887-f001:**
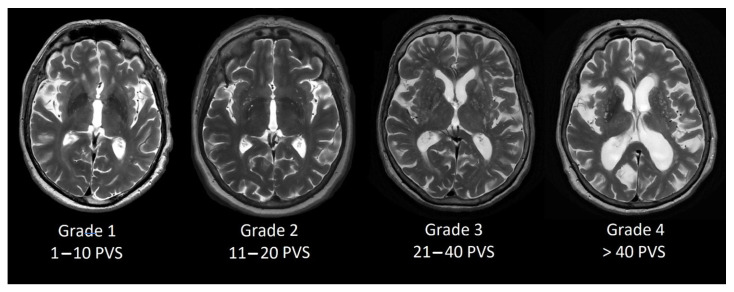
Dilated PVS in basal ganglia from Grade 1 to Grade 4.

**Figure 2 medicina-58-00887-f002:**
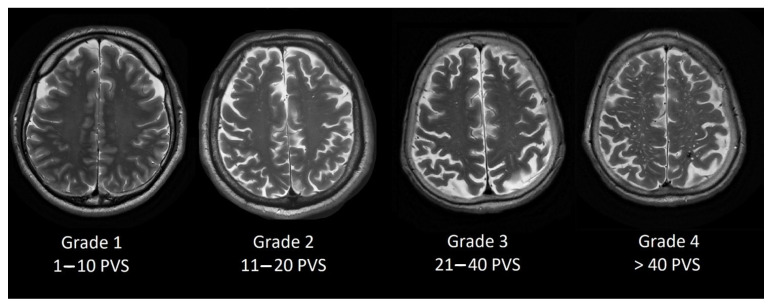
Dilated PVS in centrum semiovale from Grade 1 to Grade 4.

**Figure 3 medicina-58-00887-f003:**
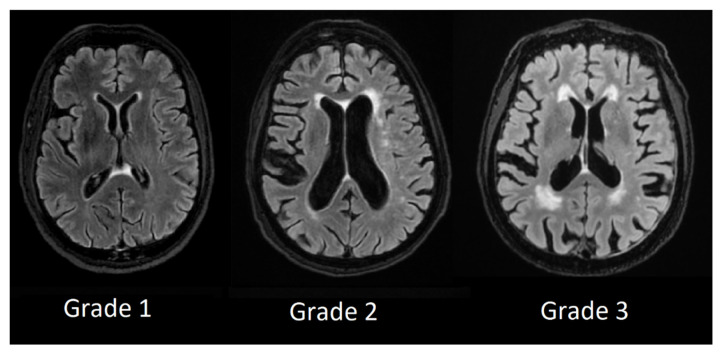
Periventricular white matter hyperintensities based on Fazekas scale from Grade 1 to Grade 3.

**Figure 4 medicina-58-00887-f004:**
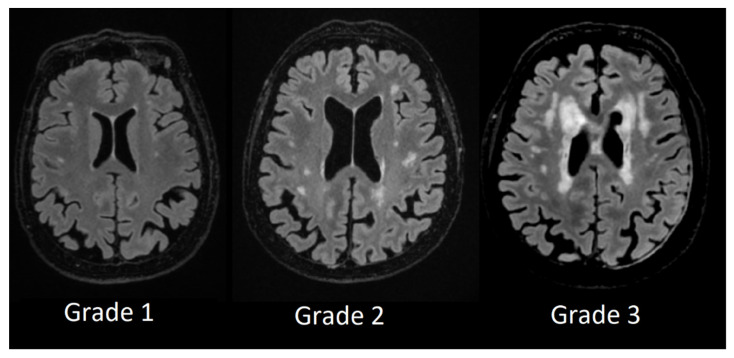
Deep white matter hyperintensities based on Fazekas scale from Grade 1 to Grade 3.

**Figure 5 medicina-58-00887-f005:**
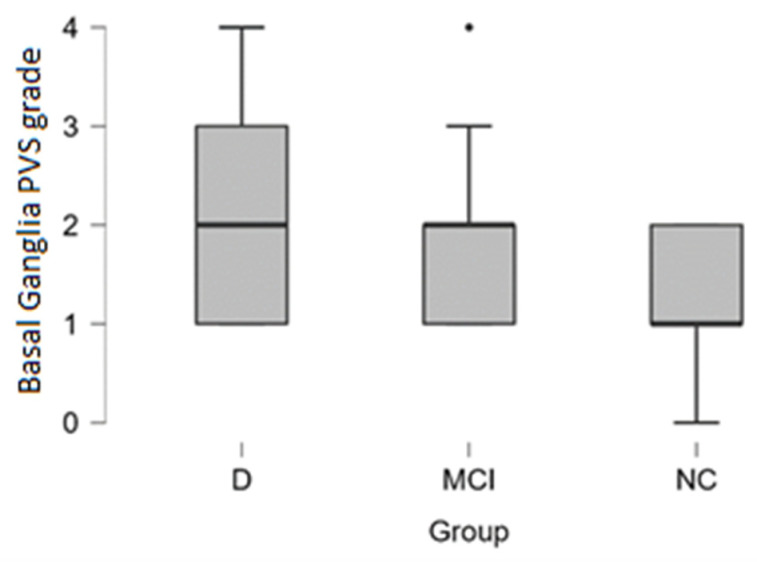
Basal ganglia PVS dilatation grade boxplot in (from left to right) dementia, MCI, and normal cognition groups (black dot, outlier).

**Figure 6 medicina-58-00887-f006:**
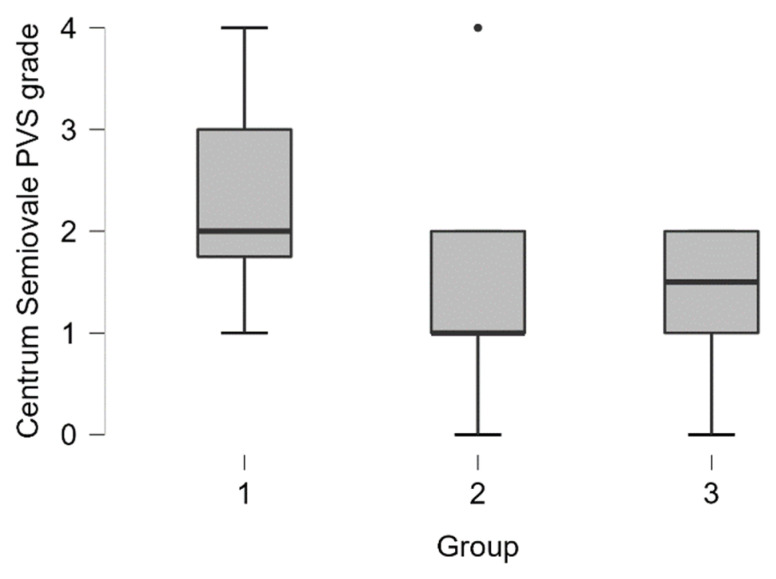
Centrum semiovale PVS dilatation grade boxplot in (from left to right) dementia, MCI, and normal cognition groups (black dot–outlier).

**Figure 7 medicina-58-00887-f007:**
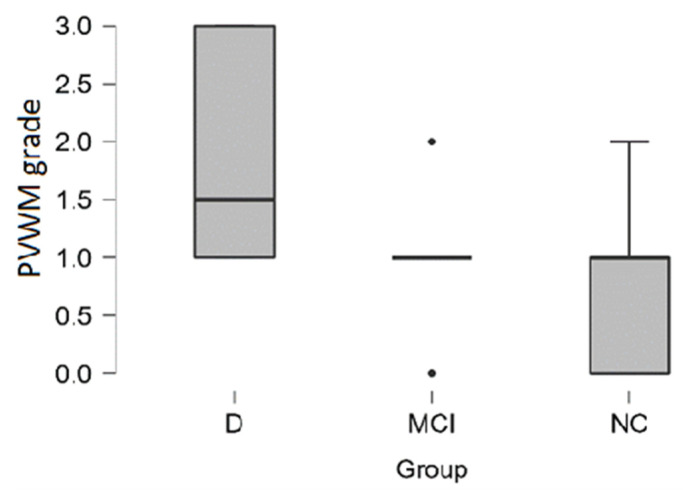
PVWM hyperintensities grade boxplot in (from left to right) dementia, MCI, and normal cognition groups (black dot–outlier).

**Figure 8 medicina-58-00887-f008:**
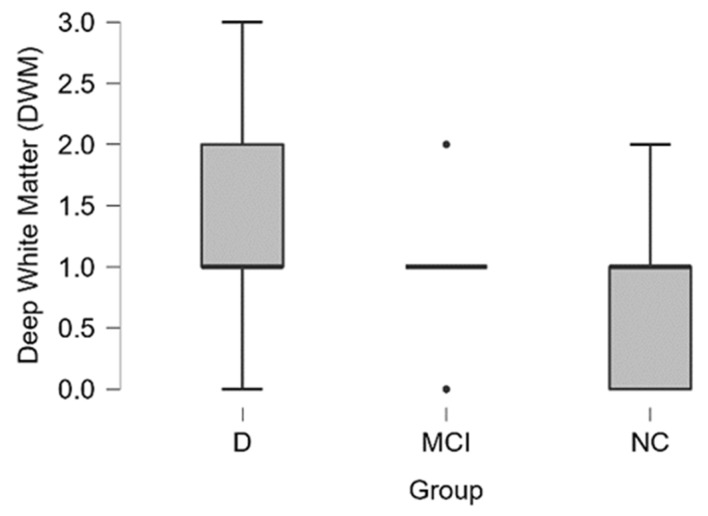
DWM hyperintensities grade boxplot in (from left to right) dementia, MCI, and normal cognition groups (black dot–outlier).

**Figure 9 medicina-58-00887-f009:**
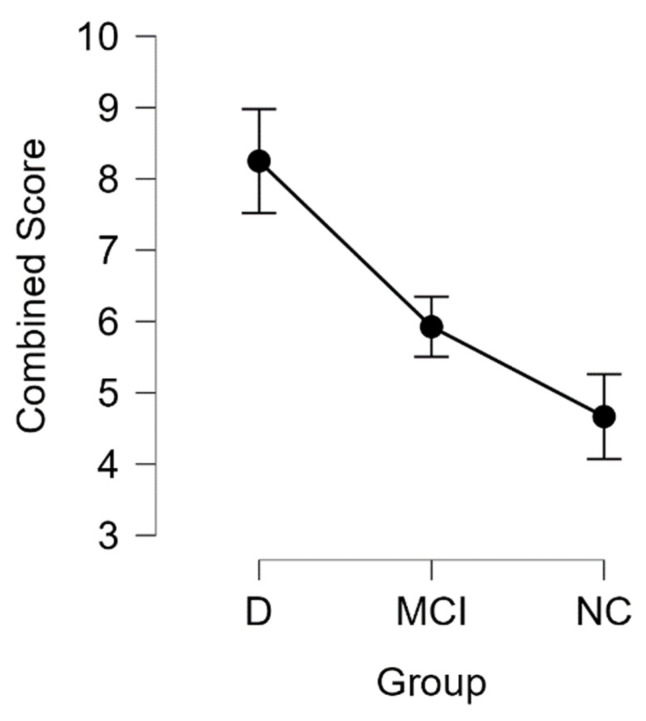
The descriptive plot of combined score in D, MCI, and NC groups with mean value and SD.

**Figure 10 medicina-58-00887-f010:**
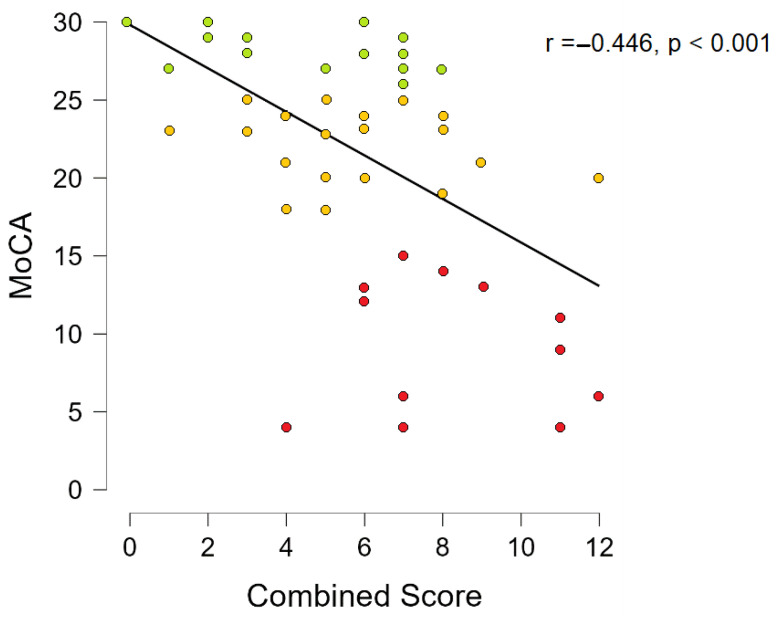
MoCA and combined score scatter plot with a trendline (red dots, participants in dementia group; yellow dots, participants in MCI group; green dots, participants in normal cognition group).

**Table 1 medicina-58-00887-t001:** Participant demographic data, gender, and MoCA results.

	Gender (F:M)			Age			MoCA		
	D	MCI	NC	D	MCI	NC	D	MCI	NC
N	7:5	19:8	15:3	12	27	18	12	27	18
Mean				73.3	72.3	63.9	9.3	22.4	28.1
Std. Deviation				10.2	7.2	15	4.2	2.4	1.3
Minimum				62	57	35	4	18	26
Maximum				96	85	83	15	25	30
χ^2^	2.291			2.633			48.016 ***		

*** = *p* < 0.001.

**Table 2 medicina-58-00887-t002:** Basal Ganglia PVS dilatation grade in dementia, MCI and normal cognition groups.

Basal Ganglia PVS Grade	Group			Total
	D	MCI	NC	
**0**	0	0	2	**2**
1	5	13	8	**26**
2	3	8	8	**19**
3	3	5	0	**8**
4	1	1	0	**2**
**Total**	**12**	**27**	**18**	**57**

**Table 3 medicina-58-00887-t003:** Centrum Semiovale PVS grade dilatation in dementia, MCI and normal cognition groups.

Centrum Semiovale PVS Grade	Group			Total
	D	MCI	NC	
**0**	0	2	2	**4**
1	3	12	7	**22**
2	4	11	9	**24**
3	3	0	0	**3**
4	2	2	0	**4**
**Total**	**12**	**27**	**18**	**57**

**Table 4 medicina-58-00887-t004:** Dunn post-Hoc comparison of Centrum semiovale PVS between D–MCI, D–NC, and MCI–NC groups.

Comparison	z	W_i_	W_j_	*p*	p_bonf_	p_holm_
**D–MCI**	**2.254**	**38.958**	26.870	0.012 *	0.036 *	0.030 *
D–NC	2.327	38.958	25.556	0.010 **	0.030 *	0.030 *
MCI–NC	0.280	26.870	25.556	0.390	1.000	0.390

* *p* < 0.05, ** *p* < 0.01.

**Table 5 medicina-58-00887-t005:** Periventricular White Matter (PVWM) hyperintensity grade in dementia, MCI and normal cognition groups.

PVWM Hyperintensities Grade	Group			Total
	D	MCI	NC	
**0**	0	4	6	**10**
1	6	18	11	**35**
2	2	5	1	**8**
3	4	0	0	**4**
**Total**	**12**	**27**	**18**	**57**

**Table 6 medicina-58-00887-t006:** Dunn post-Hoc comparison of PVWM hyperintensities between D–MCI, D–NC, and MCI–NC groups.

Comparison	z	W_i_	W_j_	*p*	p_bonf_	p_holm_
D–MCI	2.410	40.750	28.648	0.008 **	0.024 *	0.016 *
D–NC	3.533	40.750	21.694	<0.001 ***	<0.001 ***	<0.001 ***
MCI–NC	1.579	28.648	21.694	0.057	0.171	0.057

* *p* < 0.05, ** *p* < 0.01, *** *p* < 0.001.

**Table 7 medicina-58-00887-t007:** DWM hyperintensity grade in dementia, MCI and normal cognition groups.

DWM Hyperintensities Grade	Group			Total
	D	MCI	NC	
**0**	2	5	6	**13**
1	5	18	10	**33**
2	4	4	2	**10**
3	1	0	0	**1**
**Total**	**12**	**27**	**18**	**57**

**Table 8 medicina-58-00887-t008:** Combined score descriptive statistics in NC, MCI, and D groups.

Group	Mean	SD	N
D	8.250	2.527	12
MCI	5.926	2.183	27
NC	4.667	2.521	18

**Table 9 medicina-58-00887-t009:** Dunn’s Post Hoc Comparisons of Combined score between D–MCI, D–NC, and MCI–NC groups.

Comparison	z	W_i_	W_j_	*p*	p_bonf_	p_holm_
D–MCI	2.412	41.708	27.963	0.008 **	0.024 *	0.016 *
D–NC	3.205	41.708	22.083	<0.001 ***	0.002 **	0.002 **
MCI–NC	1.176	27.963	22.083	0.120	0.359	0.120

* *p* < 0.05, ** *p* < 0.01, *** *p* < 0.001.

**Table 10 medicina-58-00887-t010:** MoCA score correlation with basal ganglia PVS grade, centrum semiovale PVS grade, midbrain PVS grade, PVWM, DWM, and combined score.

Variable		MoCA
Basal Ganglia PVS grade	Spearman’s rho	−0.268
	*p*-value	0.044
Centrum Semiovale PVS grade	Spearman’s rho	−0.288
	*p*-value	0.030
Midbrain PVS grade	Spearman’s rho	−0.223
	*p*-value	0.095
Periventricular White Matter (PVWM)	Spearman’s rho	−0.523
hyperintensities	*p*-value	<0.001
Deep White Matter (DWM)	Spearman’s rho	−0.300
hyperintensities	*p*-value	0.023
Combined Score	Spearman’s rho	−0.446
	*p*-value	<0.001

## Data Availability

The datasets presented in this article are not readily available since it could have potentially identifiable data. Requests to access the datasets should be directed to nauris.zdanovskis@rsu.lv.

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
