# Peer review of "Combined Score of Perivascular Space Dilatation and White Matter Hyperintensities in Patients with Normal Cognition, Mild Cognitive Impairment, and Dementia"

_medicina, 2022, doi:10.3390/medicina58070887_

Round 1

Reviewer 1 Report

Summary:

Thank You for the opportunity to review this paper. I think that this is a very interesting study that deals effectively with an interesting issue to develop possible new radiological biomarkers. However, I think that the quality of the paper does not match its scientific soundness. The “Materials and Methods” section should be re-written to include only the description of the technique employed in the study and the results should be moved to the appropriate section. 

Title: I think that the title is a little bit too vague. You should highlight the important aspects of your work (i.e., the combined score).

Abstract: Informative in line with the main text. 

Keywords: appropriate.

Informed Consent: adequate.

Ethical Committee: adequate.

Minor Comments:

Introduction:

  1. Line 63: “Smeijer et al. 2019”. References should be standardized. 

Materials and Methods:

  1. Methods should be extensively re-written. This section should describe the methods employed to perform the study. No results should be included in this section.
  2. The design of the study (retrospective, prospective, etc..) should be described in this part.

Results:

  1. Page 182: “Results” should be the title of the next section.

Discussion:

  1. Line 325: “glymphatic”? Is this a typo?

References: satisfactory

Tables: adequate.

Figures: very interesting.

Author Response

Dear reviewer,

Thank you for your time and comments. Below are the answers to comments.

  1. “I think that the title is a little bit too vague. You should highlight the important aspects of your work (i.e., the combined score)”

After reconsideration in our group, we agree – we changed the title with an emphasis on the combined score.

  1. Line 63: “Smeijer et al. 2019”. References should be standardized.

Thank You! We fixed the issue. Now it is the same as in the rest of the document!

  1. Methods should be extensively re-written. This section should describe the methods employed to perform the study. No results should be included in this section.

We moved all the results (including demographic data and MoCA scores) to the results section. In the “Materials and Methods” section, we left only a brief description on patient selection.

  1. The design of the study (retrospective, prospective, etc..) should be described in this part.

Thank You! Included study design in the “Materials and Methods” section.

  1. Page 182: “Results” should be the title of the next section.

I have fixed the issue! Thank You!

  1. Line 325: “glymphatic”? Is this a typo?

This is not a typo. The term “glymphatic system” was introduced in the last decade. In general, it is a waste clearance system that utilizes perivascular channels, formed by astroglial cells, to promote the efficient elimination of soluble proteins and metabolites from the central nervous system (source: Jessen NA, Munk AS, Lundgaard I, Nedergaard M. The Glymphatic System: A Beginner's Guide. Neurochem Res. 2015;40(12):2583-2599. doi:10.1007/s11064-015-1581-6).

Thank you for your time and suggestions!

Have a nice day!

Best regards,

Nauris Zdanovskis

Reviewer 2 Report

The submitted manuscript entitled "Perivascular Spaces and White Matter Hyperintensities in Patients with Normal Cognition, Mild Cognitive Impairment, and Dementia" performed visual grading of perivascular space (PVS) dilation and white matter hyperintensity (WMH) in several brain regions, including basal ganglia, centrum semiovale, midbrain, deep white matter, and periventricular white matter, using magnetic resonance images for patients with normal cognition (NC), mild cognitive impairment (MCI), and dementia (D). The grading scores for PVS and WMH were compared between the three groups, and were further combined to correlate with cognitive function (MoCA). The results showed that PVS dilation of Centrum semiovalue was more severe in MCI and D groups, and had significant difference between D-MCI and D-NC pairs. Periventricular white matter was more severe in MCI and D groups, and had significant differences between D-MCI and D-NC pairs. In addition, the combined score of PVS dilation and WMH showed similar results to the PVS dilation and WMH, but had singificant correlation with MoCA scores. Although the study showed some interesting findings, the novelty and methodology are the main weakness and some addtional analysis should be substantially improved. 

1) In abstract, the abstract should be structured as purpose, materials and methods, results, and conclusion. However, the conclusion part is missing and needs to be added.

2) Lines 91-93, the hypothesis and purpose are not well addressed. Statements regarding the novelty of this study should be added to enhance the contribution of this study.

3) Lines 110-112, the enrolled patients are inhomogeneous, it is unclear to me whether or not different subtypes of dementia could lead to different results.

4) Line 123, the table is not concise, and the values should be expressed as mean +/- standard deviation.

5) Lines 124-125, the statement should be moved to the subsection of statistical analysis.

6) Lines 126-131, the statements should be moved to the results section.

7) Lines 133-137, the abbreviations should be defined or spelled out at first use.

8) In methods, the grading of PVS dilation and WMH should be performed by at least two independent neuroradiologists, and inter-observer variability or intra-class correlation should be analyzed to show the reproducibility of the subjective grading.

9) Line 182, "3. Results" should be moved to the next line.

10) Figure 10, the correlation was performed only between the combined score and MoCA. The correlation should be analyzed for PVS dilation and WMH with MoCA respectively, in order to enhance the usefulness of the combined score.

11) In results, the comparisons of PVS dilation and WMH scores between the three groups are similar to the combined score. It is not clear to me whether the combined score is superior to traditional scores (PVS dilation and WMH). Further analysis is necesary.

12) The enrolled NC groups were youner than other groups. Hence, it is difficult to evaluate the comparison of PVS dilation, WMH, and the combined scores between NC and other groups.

Author Response

Dear reviewer,

Thank You for Your time and comments. Below are the answers to the comments.

1) In the abstract, the abstract should be structured as purpose, materials and methods, results, and conclusion. However, the conclusion part is missing and needs to be added.

Thank You! We restructured the abstract part and added the conclusion part too.

2) Lines 91-93, the hypothesis and purpose are not well addressed. Statements regarding the novelty of this study should be added to enhance the contribution of this study.

We added statements and emphasized that at the moment PVS space dilatation evaluation is not widely recognized nor endorsed. Also added that we analyzed the combined score of PVS dilatation and WMH evaluation.

3) In Lines 110-112, the enrolled patients are inhomogeneous, it is unclear to me whether different subtypes of dementia could lead to different results.

We did not examine a specific type of dementia. Based on the neurological assessment and magnetic resonance imaging data, participants in the MCI and D groups had mixed type dementia (i.e., showing signs of at least two different types of dementia, in most cases vascular and Alzheimer’s).

4) In Line 123, the table is not concise, and the values should be expressed as mean +/- standard deviation.

The mean values and standard deviation are mentioned in the table. We tend to avoid using “±” sign because in that case, it is unclear whether it is “standard deviation” or “standard error”.

5) Lines 124-125, the statement should be moved to the subsection of statistical analysis. Lines 126-131, the statements should be moved to the results section.

We moved the lines to the results section. Thank You!

7) Lines 133-137, the abbreviations should be defined or spelled out at first use.

We spelled out MR technical abbreviations. Thank You!

8) In methods, the grading of PVS dilation and WMH should be performed by at least two independent neuroradiologists, and inter-observer variability or intra-class correlation should be analyzed to show the reproducibility of the subjective grading.

In our study, we did not assess inter-observer variability or inter-class correlation. Included figures with visual scales could help with the reproducibility of the study in the future.

9) Line 182, "3. Results" should be moved to the next line.

Thank You! Fixed it!

10) Figure 10, the correlation was performed only between the combined score and MoCA. The correlation should be analyzed for PVS dilation and WMH with MoCA respectively, in order to enhance the usefulness of the combined score.

We added correlation analysis for PVS and WMH with MoCA in the results section.

11) In results, the comparisons of PVS dilation and WMH scores between the three groups are similar to the combined score. It is not clear to me whether the combined score is superior to traditional scores (PVS dilation and WMH). Further analysis is necessary.

In our study, we are not stating that a combined score is superior to traditional scores. We summarize our findings in regard to PVS dilatation in 3 different regions and WMH in 2 regions. The combined score is an additional measure that we used to summarize our findings in regard to PVS dilatation and WMH.  

12) The enrolled NC groups were younger than other groups. Hence, it is difficult to evaluate the comparison of PVS dilation, WMH, and the combined scores between NC and other groups.

Even though NC group was younger, when correcting the combined score for age and gender statistically significant results remained. Also, a Kruskal-Wallis H test was conducted to examine the age differences between groups, and no significant differences were found between the groups (χ2 = 2.633, p = 0.268).

Additionally, we did an extensive English recheck with WriteFull, Grammarly, and academic editor.

Thank you for your time and suggestions!

Have a nice day!

Best regards,

Nauris Zdanovskis

Reviewer 3 Report

Well written introduction.

Clear the scientific background.

The authors have briefly described the definition of cerebral perivascular spaces (PVS) or Virchow-Robin spaces and White matter hyperintensities (WMH), inserting appropriate recent bibliographic references and explaining why thy could be used and identified as MRI biomarkers  for cerebral microvascular disease.

Adequate description of the inclusion and exclusion criteria.

Good analysis of the MR images to classify PVS and WMI according to the most recent scientific papers published on the topic.

Adequate description of the inclusion and exclusion criteria.

Rigorous statistical analysis.

The reflections in the discussion are noteworthy and interesting.

Adequate description of the study limitations. 

The major strength of this study is to derive a predictive assessment of the risk of mild cognitive impairment from two conventional MRI features assessments: PVS and WMI.

The proposed assessments could guide a more careful analysis of MRI exams in daily activity to identify people most at risk of converting to AD (Alzheimer’s Disease).

Author Response

Dear reviewer,

Thank you for your time and comments.

To improve English we did an extensive English recheck with WriteFull, Grammarly, and academic editor.

Have a nice day!

Best regards,

Nauris Zdanovskis

Round 2

Reviewer 2 Report

Lines 53-54, PVWH should be removed because it was not mentioned in the following text.

Line 100, perivascular space should be replaced by PVS, and white matter hyperintensities by WMH.

Line 174, authors should indicate which method was used in the correlation analysis.

Line 335, there is no need to define the abbreviations for PVWM and DWM again.

Line 337, table 11 should be corrected to table 10.

Lines 355-356, references should be cited for previous studies.

Lines 399-403, the objective here is not parallel to the objective of this study. Some statements are repeated in Lines 415-419. 

Author Response

Dear reviewer,

Thank You for Your revised comments and additional suggestions. Below are the corrections.

  1. Lines 53-54, PVWH should be removed because it was not mentioned in the following text.

Removed! Thank You!

  1. Line 100, perivascular space should be replaced by PVS, and white matter hyperintensities by WMH.

Changed respective words to PVS and WMH. Thank You!

  1. Line 174, authors should indicate which method was used in the correlation analysis.

Added a sentence mentioning Spearman’s correlation.

  1. Line 335, there is no need to define the abbreviations for PVWM and DWM again.

 We took out the definitions of these abbreviations. Thank You!

  1. Line 337, table 11 should be corrected to table 10.

Corrected!

  1. Lines 355-356, references should be cited for previous studies.

 We added comments regarding previous research and added citations from previous studies!

  1. Lines 399-403, the objective here is not parallel to the objective of this study. Some statements are repeated in Lines 415-419. 

We added more information in lines 399-403 to better align with conclusions and the objective of the study.

Thank you for your time and suggestions!

Have a nice day!

Best regards,

Nauris Zdanovskis

This manuscript is a resubmission of an earlier submission. The following is a list of the peer review reports and author responses from that submission.

Round 1

Reviewer 1 Report

  1. On line 30 of the abstract, …between the groups…, what are the groups?
  2. The number of subjects in each group is too small and may not have valid statistical power.
  3. "Cerebral small vessel disease and the risk of dementia: A systematic review and meta-analysis of population-based evidence." published in 2018, "White matter hyperintensities and risks of cognitive impairment and dementia: A systematic review and meta-analysis of 36 prospective studies.” published in 2021,and “ WMH were associated with increased risk of cognitive dysfunction and could become a neuroimaging indicator of dementia”, have been used to evaluate the relationship between WHM and the cognitive function in AD disease spectrum. And conclusions are drawn. So what is the innovation of the author's work? Please explain.
  4. actually, cerebral small vessel disease includes WMH, microinfarcts, perivascular spaces and microbleeds. Why did the authors only study WMH and perivascular space in this study, and why did not include indicators such as microinfarcts and microbleeds. Please explain.
  5. Given that WMH and Perivascular spaces belong to the category of cerebral small vessel disease, they may be related to high cerebrovascular factors. Therefore, clinical characteristics such as hypertension, diabetes, BMI, ApoE gene, smoking, drinking, blood pressure level, and education level need to be included and comparison between groups. Although the author mentions this in the limits, it needs to be clarified in this study.
  6. It is not convincing that the NC, MCI and D groups rely solely on MOCA for grouping, and the authors did not even use the CDR, which is an important indicator for distinguishing MCI and D. In fact, since NIA-AA released a new diagnostic framework for AD continuum disease spectrum in 2018, more and more studies need to group AD disease spectrum according to the A/T/N framework.
  7. Table 1 needs to be concise, it seems too verbose.
  8. The description of Statistical analysis is not comprehensive. Which one is used for post hoc analysis? Which method was used in the correlation analysis and whether it was corrected, and did the correlation analysis include indicators such as age, gender, and education level as covariates? None of this is clear and needs to be described.
  9. In Figure 10, why did the author put the three groups together for a Pearson correlation analysis, which would confound the results with some confounding factors. What is the basis? In addition, the authors were asked to conduct correlation analysis within the group and report the results. And please use a graphic to mark each disease group represented by each color in the picture.
  10. Do WMH and Perivascular spaces influence each other? Whether there is an interaction between the two effects on cognition needs to be elucidated by the authors.
  11. In the Discussion, the author's discussion is too superficial, and does not integrate WMH and Perivascular spaces with the cognition, pathology, etc. of AD disease spectrum.
  12. Similarly, the conclusion is too simplistic, it seems that it is only listing the results, and there is no effective summary of the full text.

Reviewer 2 Report

The authors explored potential differences in WMH severity and quantity of enlarged PVS (ePVS) between cognitively normal, MCI, and demented groups.  There has been some interest in this area in trying to use MRI-derived biomarkers for early detection of cerebrovascular diseases and potential subsequent cognitive impairment.  However, I must recommend rejection for the current manuscript as I have multiple significant methodological concerns. To be fair, the authors do bring up some of these methodological limitations in the discussion section.

Major Comments

1) The authors make several claims in the results sections (line 184, line 219, line 250) which are simply not supported by statistics, even if it is true by eye.  These sentences will need to be rephrased so that they are not claiming to represent statistically significant results.  In addition, it is good practice to at least put the p-value, when applicable, at the end of the statement for clarity.

2) The Methods section is sparse enough that the current study cannot really be replicated. While section 2.1 is reported in sufficient detail for the most part for this reviewer, the rest of the sections seem to be missing important details. In particular, it would be helpful to have more information about scan parameters (section 2.2) particularly voxel size, matrix size, did the scan cover the whole brain etc. Did the authors evaluate PVS grading on a particular slice or slices, and what orientation were the images in (sagittal, axial)? Why was midbrain PVS treated differently than basal ganglia or centrum semiovale? How was lesion location divided into periventricular and deep locations?  What was the threshold? Finally, statistical analysis could be described further, for instance, a regression analysis was conducted but it is not mentioned in section 2.4.  Would also be useful to know about outliers, normality, etc. about the data as well.

3) As the authors mentioned (line 342), the participants in the healthy group were younger (~10 years) than participants in the other 2 groups.  But in addition, the healthy group also had a greater proportion of females (Table 1), while the MCI and demented groups were associated with sort of non-specific dementias. The N is also fairly low (as mentioned by authors, line 344), which is accentuated when the sample was further divided into 3 groups. Taken collectively, and given that these factors cannot be corrected by the study design, it is very difficult to make definitive conclusions from the present study.

4) While I see that composite scores accounting for ePVS and WMH severity in several regions were calculated (sections 2.3, 3.3) there seems to be little basis for this calculation.  Is there literature supporting its validity?  Why is PVS space dilation in midbrain worth three times less than the other two regions?  Using the current weighting, PVS space has slightly more weight than WMH volume, is this appropriate?

Minor comments 

  1. WMHs and ePVS are referred to as "direct and indirect" imaging biomarkers of cerebral microvascular integrity (lines 17, 44, 87).  I am not sure I understand. As these are MRI biomarkers I would consider them indirect biomarkers.
  2. ePVS were ultimately counted in the current study.  If this is the case, the language of PVS dilation can be confusing in places (like the discussion) because it is not clear if the authors are referring to the magnitude of dilation (i.e. “bigger” PVS) or more likely that they are referring to a greater quantify of PVS, which is what is measured in the current study. 
  3. I do not understand what is meant in a sentence like “In the dementia group, the average grade of DWM hyperintensities had a higher value according to the Fazekas scale and a higher grade was encountered more often in the dementia group than in other groups (line 250)”. What is the first part of the sentence trying to communicate? Had a higher value according to the Fazekas scale than what?
  4. I believe that in line 260 only 5 out of 12 participants had grade 2 or grade 3 hyperintensities of DWM in the dementia group, according to Table 7.
  5. I believe that the discussion could be more focused, i.e. explicitly state what is gained by combining WMH volume and PVS information in regards to clinical outcomes and implications for the field.

Reviewer 3 Report

As PVS and WMH have important role in cognition, your attempt to evaluate them as biomarkers is meaningful.

However, I would like to identify a number of points that may undermine the results.

1. In line 103, All participants had at least 16 years of higher education. 

 —> Is there any reason why all participants’ education level is extremely high? It can cause significant biases. Because people with large cognitive reserve can show a large gap between structural changes in the brain and actual cognitive function. You classified groups of participants by cognitive fx.(MoCA), the results may be influenced by biased participants.

2. In line 108, Based on neurological 108 assessment and magnetic resonance imaging data, participants in the MCI and D groups 109 were not exclusively classified as one type of dementia

—>I can undersand you could not differentiate the types of dementia accurately, but I wonder if you could evaluate some structural changes (e.g. hippocampal atrophy, certain cerebral cortex atrophy). It will enhance your results. 

3. In line 201, Table 3 

—> Only Table 3. has different name on the X-axis. (Group name)

4. In line 306, there were statistically significant differences in PVWM between our groups and there were no statistically significant differences in the DWM group. Therefore, our results are consistent with Kim et al., 2008, and contradict Bolandzadeh et al., 2012. More studies are needed to confirm our findings. 

—> Your results may be meaningful just by replicating previous studies, but it is advisable to explore your own opinions supported by appropriate evidence.

Please consider to edit manuscript as much as possible according to my recommendations.

And If you cannot edit it, it would be nice if you could remark the limitations.